

# Controlled release of methyl salicylate by biosorbents delays the ripening of banana fruit

Chalida Cholmaitri[1], Apiradee Uthairatanakij[1], Natta Laohakunjit[2], Pongphen Jitareerat[1] and Withawat Mingvanish[3]

[1] Postharvest Technology, King Mongkut's University of Technology Thonburi, Bangkok, Thailand
[2] Biochemical Technology, King Mongkut's University of Technology Thonburi, Bangkok, Thailand
[3] Department of Chemistry, King Mongkut's University of Technology Thonburi, Bangkok, Thailand

## ABSTRACT

The efficiencies of rice flour (RF) and rice husk (RH) as agents of the controlled release of methyl salicylate (RF-MeSA and RH-MeSA, respectively) were investigated. The adsorption percentage of RH-MeSA was significantly higher (two-fold) than that of RF-MeSA owing to its higher specific surface area and total pore volume. However, both materials are classified as mesoporous materials. Scanning electron microscopy, X-ray diffraction, Fourier transform infrared spectroscopy and thermogravimetric analysis confirmed that MeSA diffused toward the pores and covered the surfaces of RF and RH. A temperature increase from 25 °C to 40 °C and an increase in relative humidity from 75% to 95% stimulated the release of MeSA. The kinetically controlled release of RF-MeSA and RH-MeSA was in line with a Fickian diffusion mechanism. Both RF-MeSA and RH-MeSA significantly delayed the ripening of banana fruit compared to the control. The results indicate that RF and RH can be used as biosorbent materials for the adsorption and controlled release of MeSA without chemical and mechanical modification.

## INTRODUCTION

Physiological and biochemical changes during fruit ripening lead to changes in color, a peak in respiration, a burst in ethylene production, softening and declines in acidity (*Gray et al., 1994*). Ethylene is an endogenous plant hormone responsible for fruit ripening and senescence (*Pratt & Goeschl, 1969*). Ethylene is synthesized from methionine through the intermediaries S-adenosyl methionine (SAM) and 1-aminocyclopropane-1-carboxylic acid (ACC). The enzyme that converts methionine to SAM is SAM synthase, while ACC synthase converts SAM to ACC and ACC oxidase catalyzes the oxidation of ACC to ethylene (*Yang & Hoffman, 1984*).

Corresponding author
Apiradee Uthairatanakij,
apiradee.uth@kmutt.ac.th

Methyl salicylate (MeSA) is a volatile organic compound that plays an important role in inhibiting the activity of ACC synthase and ACC oxidase in tomatoes and plums (*Ding et al., 2002*; *Khan, Singh & Abbasi, 2007*). It delays the postharvest ripening process in papayas, mangos and sweet peppers (*González Aguilar, Buta & Wang, 2003*; *Fung et al., 2004*; *Han et al., 2006*). Moreover, it has a strong ability to inhibit fungal infections and reduce chilling injury symptoms (*Ryals et al., 1996*; *Fung et al., 2006*; *Min et al., 2018*). MeSA is used as a food additive that is classified by the U.S. Food and Drug Administration as a Generally Recognized as Safe (GRAS) substance (*Chanjirakul et al., 2006*). Fumigation with 1.0 mmolL$^{-1}$ of MeSA vapor for 16 h at 20 °C reduced the weight loss, respiration rate, softening and total acidity of sweet cherries (*Giménez et al., 2016*). MeSA vapor at 0.5 mmolL$^{-1}$ for 16 h at 23 °C prevented red color development, ethylene production and respiration in tomatoes (*Ding & Wang, 2003*). Moreover, pomegranates dipped in 1.0 mmolL$^{-1}$ MeSA showed a significant reduction in chilling injury and maintained quality and health when stored at 2 °C for 3 months (*Sayyari et al., 2011*). *Chotikakham et al. (2019)* found that bananas dipped in MeSA solution at concentrations of 2 and 4 mmolL$^{-1}$ maintained fruit quality and reduced peel spotting.

Methyl salicylate is usually applied to fruit either by fumigation or by dipping them in a solution (*Srivastava & Dwivedi, 2000*; *Bagnato et al., 2003*). However, these applications are limited in their efficacy because of the rapid release of this volatile organic compound (*Mir et al., 2004*). A controlled release system that can slow the release of active compounds has been applied to prolong shelf life and control the quality of meat, fish, poultry, bread, cheese, beverages, fruit and vegetables (*López-Rubio et al., 2004*). Factors affecting controlled release include the coating materials (pore size, wall thickness, and coating layers), type of active compound or core, and the release environment (temperature and relative humidity) (*Yebra, Kiil & Dam-Johansen, 2004*; *Hoffman, 2008*; *Vilar, Tulla-Puche & Albeicio, 2012*).

A number of biosorbent materials, such as α-cyclodextrin (α-CD), β-cyclodextrin (β-CD), activated carbon and zeolite are used in controlled release techniques (*Kamada et al., 2002*; *Hamadi, Swaminathan & Chen, 2004*; *Fernández-Pérez et al., 2005*; *Namazi, Bahrami & Entezami, 2005*; *Trichard et al., 2007*; *Fatouros et al., 2011*). A 1-methylcyclopropene (1-MCP)/α-CD inclusion complex powder has been shown to delay the ripening and prolong the shelf life of tomato (*Ariyanto, Bercis & Yoshii, 2019*). β-CD inclusion with hexanal has been shown to inhibit postharvest disease in berries (*Almenar et al., 2007*). In addition, activated carbon with ethanol has been shown to prevent microbial contamination of strawberries (*Kawagoe, Takikawa & Hikosaka, 1998*). Zeolite is used as a fertilizer carrier to control the release of nitrate (*Li, 2003*), phosphate (*Bansiwal et al., 2006*) and sulfate (*Li & Zhang, 2010*). Cyclodextrins (CDs), which are produced during the degradation of starch by enzymatic conversion, are water-soluble, with a hydrophobic interior and hydrophilic exterior (*Cevallos, Buera & Elizalde, 2010*). They are used to coat or encapsulate insoluble compounds (*Teranishi & Shimomura, 2014*). However, biosorbent materials that are used to slow the ripening process and reduce postharvest loss need to be water-insoluble. For example, cellulose is a natural biopolymer that is water-insoluble, but with a highly charged and, thus, modifiable surface

(*Bochek, 2003*). Therefore, the use of agricultural by-products and wastes as biosorbent materials is becoming an area of interest because they are renewable and low-cost. The characteristics of various types of starches, such as a double-helix of linear amylose molecules have the ability to bind with other hydrophobic molecules such as iodine, fatty acids, or aromatic compounds, making them potential candidates for applications related to adsorption and controlled release (*Geera et al., 2006*). Potato and corn starch can adsorb volatile compounds (*BeMiller & Pratt, 1981*), and amylose extracted from potato starch can bind with *n*-butyl alcohol, iso-amyl alcohol, menthone, and other compounds (*Takeo & Kuge, 1969*). Moreover, rice starch has been used as an ingredient in dry shampoo because it absorbs oil in the hair (*Santander-Ortega et al., 2010*). Assam Bora rice starch can be used as a carrier for controlled-release drug delivery (*Ahmad et al., 2012*). Rice starch-konjac glucomannan (KGM) blended films with MeSA have shown potential as agents of controlled release of bioactive compounds (*Satirapipathkul & Meesukanun, 2013*).

Rice husk (RH), an agricultural waste product that is generated in the milling process of rice grain, is normally used as feed for livestock. The major constituents of RH are cellulose (35%), lignin (25%), silica (20%), crude protein (3%) and ash (17%) (*Ugheoke & Mamat, 2012*). The latter constituent imbues the husks with high surface area and porosity, two important attributes that facilitate adsorption and desorption (*Basha et al., 2005*). Rice husk ash is a source of silica, which is used in slow-release drug delivery systems (*Prawingwong et al., 2009*). Urea coated with rice husk charcoal has the potential to slow the release of nitrogen fertilizer (*Xiaoyu et al., 2013*). To the best of our knowledge, there are no reports in the literature that examine the applicability of rice flour (RF) and RH to the adsorption and controlled release of MeSA. Therefore, the objectives of this study were to investigate the efficacy of RF and RH without a modified surface in adsorbing MeSA and their ability to slowly release this compound, as well as to understand the release kinetics of these two biosorbents. RF and RH biosorbents were used to examine the delay in fruit ripening, using bananas as a model. Bananas are a climacteric fruit which show a sharp increase in ethylene production with a high respiration rate during the time of ripening (*Bouzayen et al., 2010*). These two biosorbents can potentially be applied in postharvest technology as an environmentally friendly method of delaying fruit ripening.

## MATERIALS AND METHODS

### Preparation and characterization of biosorbents

#### Materials

Rice flour, derived from broken rice (*Oryza sativa* L.), was purchased from a local market in Ratchaburi Province, Thailand. The amylose content was 30.2% and the moisture content was 10.1%. Rice husk was supplied by Nugreen Co., Ltd. (Thailand), and had a moisture content of 7.6%. Moisture contents were determined following the method of the Association of Official Analytical Chemists (*Association of Official Analytical Chemists (AOAC), 2000*). The RF and RH were ground into powder using a grinder (Cuisinart SG-10 HK, China), then dried at 60 °C for 96 h in a hot air oven to remove moisture.

The powders were passed through a 0.250–0.177 mm sieve and kept in a desiccator until they were used for physical characterization and study of adsorption and desorption properties. Methyl salicylate (MeSA; purity ≥ 99%) was obtained from Sigma–Aldrich (St. Louis, MO, USA).

## Measurement of Brunauer–Emmett–Teller surface areas

The specific surface area, total pore volume, and average pore diameter of RF and RH powder (0.250–0.177 mm) were determined by nitrogen sorption at 77 K using a Brunauer–Emmett–Teller (BET) surface analyzer (Model Quantachrome, Autosorb 1, Boynton Beach, Florida, USA). The surface area was determined according to the BET equation (*Brunauer, Emmett & Teller, 1983*).

## Adsorption of rice flour-methyl salicylate and rice husk-methyl salicylate

The biosorbents (RF and RH) were weighed and placed into 10 mL airtight glass vials. MeSA was added to each of the biosorbents at a ratio of 2:1 (w/w; MeSA: biosorbent) and left at 25 °C for 24 h. The adsorption percentage and physical characteristics of rice flour-methyl salicylate (RF-MeSA) and rice husk-methyl salicylate (RH-MeSA) were then assessed.

## Adsorption percentage

The RF-MeSA and RH-MeSA mixtures were filtered using Whatman® filter paper No. 1 to remove excess MeSA (*Mulugeta & Lelisa, 2014*) and weighed at 0, 1, 2, 3, 6, 12 and 24 h to determine the adsorption percentage of RF-MeSA and RH-MeSA. The adsorption percentage was calculated as follows:

$$\text{Adsorption}(\%) = [(W_i - W_t)/W_i] \times 100 \qquad (1)$$

where $W_i$ is the adsorbed weight at the initial time (g) and $W_t$ is the adsorbed weight at time $t$ (g).

## Scanning electron microscopy and energy dispersive X-ray spectroscopy

Samples of RF, RH, RF-MeSA and RH-MeSA were mounted on stubs using adhesive carbon tape and coated with gold. The morphology was investigated using a scanning electron microscope Scanning electron microscopy–energy dispersive X-ray spectroscopy (SEM–EDS; JSM 6610 LV, JEOL Ltd., Tokyo, Japan) at 2,500×$g$ magnification and an accelerating voltage of 10 kV.

## X-ray diffraction

The X-ray patterns of RF, RH, RF-MeSA and RH-MeSA samples (0.5 g) were analyzed using an X-ray diffraction (XRD) analyzer (Bruker AXS, Model D8 Discover, Billerica, MA, USA) with copper radiation at a voltage of 40 kV and 40 mA. The RF, RH, RF-MeSA and RH-MeSA samples were scanned between 2θ = 5°–60° with a scanning speed of 2° min$^{-1}$.

## Fourier transform infrared spectroscopy

For functional group analysis, 2 mg samples of RF, RH, RF-MeSA and RH-MeSA were mixed with 100 mg of potassium bromide powder; the mixture was compressed at 10 psi. The pellets were transferred into a Fourier transform infrared (FTIR) spectrometer (Perkin Elmer, Spectrum One, USA), and their spectra were recorded at a resolution of 4 $cm^{-1}$ in a range of 400–4,000 $cm^{-1}$.

## Thermogravimetric analysis

To determine thermal decomposition of RF, RH, RF-MeSA and RH-MeSA, 5 mg samples were scanned using a thermogravimetric analysis (TGA) analyzer (Perkin-Elmer, Model Pyris Diamond, USA). The samples were heated from 30 °C to 800 °C, with a temperature ramp rate of 10 °C $min^{-1}$. Nitrogen was used as the purge gas at a flow rate of 10 mL $min^{-1}$.

## Desorption of RF-MeSA and RH-MeSA

The biosorbents (RF and RH) were weighed and placed into 10 mL glass vials with aluminum caps. MeSA was added to each of the biosorbents at a ratio of 2:1 (w/w; MeSA: biosorbent) and left at 25 °C for 24 h. Samples of RF-MeSA and RH-MeSA (1.0 g) were placed into 10 mL airtight glass vials and used to study the effect of temperature and relative humidity on desorption processes.

## Effect of temperatures on desorption

Samples of RF-MeSA and RH-MeSA (1.0 g) in 10 mL airtight glass vials were heated using a heating box (Gemmy DB-006E, Taipei, Taiwan) at 25 °C and 40 °C respectively, at 70–75% relative humidity. Methyl salicylate gas (five mL) from the headspace was assayed at 0, 1, 2, 3, 6, 9, 12 and 24 h after heating commenced by gas chromatography (GC; GC-14B Shimadzu, Japan). The GC was equipped with a DB-5 column (30 m × 0.250 mm) and a flame ionization detector. The column temperature was 50 °C for 5 min, then heated to 130 °C at 12 °C $min^{-1}$, and then increased to 200 °C at 15 °C $min^{-1}$. Helium was used as the carrier gas at a flow rate of 1.0 mL $min^{-1}$. Injection and detector temperatures were set at 250 °C and 240 °C, respectively. Methyl salicylate was used as a standard compound. The percentage desorption of MeSA was calculated as follows:

$$Desorption(\%) = A_t/A_{eq} \times 100 \qquad (2)$$

where $A_t$ is the peak area of MeSA at time $t$ and $A_{eq}$ is peak area of MeSA at equilibrium.

## Effect of relative humidity on desorption

Samples of RF-MeSA and RH-MeSA (1.0 g) in 10 mL glass vials were placed in desiccators that were pre-equilibrated with saturated solution sodium chloride and potassium sulfate to attain relative humidities of 75% and 95%, respectively, at 25 °C (*Greenspan, 1977*). The equilibrium time for the samples in the desiccator was 1 h. Headspace gas (five mL) was collected in 1 min intervals and removed at 0, 1, 2, 3, 6, 9, 12 and 24 h for the analysis of MeSA gas by GC.

## Release kinetics

The release kinetics of MeSA from RF-MeSA and RH-MeSA were investigated according to the methods of *Ho, Joyce & Bhandari (2011)* with some modifications.

The Korsmeyer–Peppas model (*Korsmeyer et al., 1983*) was employed to describe the release kinetics by Eq. (3).

$$M_t/M = kt^n \tag{3}$$

where $M_t$ and $M_\infty$ represent MeSA released at time t and at equilibrium, respectively; $k$ is the rate constant; and $n$ is the release exponent calculated from the slope of the straight line. Controlled release was further evaluated by other models. First, the data were assessed as a zero-order reaction (Eq. (4)), as the cumulative amount of MeSA released vs. time:

$$C = k_0 t \tag{4}$$

where $k_0$ is the zero-order rate constant expressed in units of concentration/time and t is the time in hours. Second, the data were assessed as a first order reaction (Eq. (5)) as the log cumulative percentage of MeSA remaining vs. time:

$$\log C = \log C_0 - kt/2.303 \tag{5}$$

where $C_0$ is the initial concentration of MeSA released, $k$ is the first-order constant, and t is the time in hours. Lastly, the data were assessed using the Higuchi model (Eq. (6)) as the cumulative percentage of MeSA released vs. square root of time:

$$Q = kt^{1/2} \tag{6}$$

where $k$ is the rate constant and $t$ is the time in hours.

## Application of RF-MeSA and RH-MeSA on banana fruit

Banana (*Musa sapientum* L. "Namwa") fruit that were free of physical damage and symptoms of disease, and were at 90 d after bloom, were collected from commercial orchards located in Ratchaburi Province, Thailand. Banana hands were cut into individual fingers, washed with tap water, dipped for 5 min in sodium-hypochlorite (200 mg L$^{-1}$), and air-dried at 25 ± 2 °C before treatment. For each treatment, two fruit were placed inside a perforated polypropylene (PP) tray (4 holes, 2.0 mm perforations), with 1.0 g of RF-MeSA or RH-MeSA in Whatman$^®$ filter paper No. 1; a group without any biosorbent was used as a control. The storage period was 5 d at 25 ± 2 °C and at a relative humidity of 75 %. The samples were withdrawn for analysis at 0, 1, 3 and 5 d after initiation of the storage period. Six replicates were conducted per treatment, with two fruit per package.

Ethylene production and respiration rate were determined by placing one banana fruit in a 600 mL air-tight plastic container and incubating at 25 ± 2 °C for 1 h. A total of 1.0 mL sample of headspace gas was withdrawn and injected into the gas chromatograph (GC-2014B, Shimadzu, Japan). The ethylene production is expressed as ng kg$^{-1}$ s$^{-1}$ and the respiration rate is expressed as μg kg$^{-1}$ s$^{-1}$.

**Table 1  BET surface area of rice flour (RF) and rice husk (RH).**

| Parameters | Biosorbents | |
|---|---|---|
| | RF | RH |
| Specific surface area ($m^2\ g^{-1}$) | 2.850 ± 0.002 | 4.240 ± 0.003 |
| Total pore volume ($cm^3\ g^{-1}$) | 0.0043 ± 0.0021 | 0.0051 ± 0.0027 |
| Average pore diameter (nm) | 5.997 ± 1.428 | 4.824 ± 1.485 |

The color change of the banana peel was measured in three locations on each fruit using a colorimeter (CR-300, Minolta, Tokyo, Japan). The measurements were expressed as yellowness ($b^*$ value). Fruit firmness was measured by a texture analyzer (TA-XT plus, Stable Micro Systems, UK), and the results were expressed as force (N).

The half-life ($t_{1/2}$) release, which indicated the time it took for 50 % of the active compound to release from the biosorbent, was determined by Eq. (7) (Ho, Joyce & Bhandari, 2011)

$$t_{1/2} = \exp(-(\ln k + \ln(0.5))/n) \tag{7}$$

where $k$ is the rate constant and $n$ is the release exponent calculated from the slope.

## Statistical analysis

All experiments were arranged in a randomized complete block design (RCBD) with three replicates. The data were analyzed by analysis of variance (ANOVA) using SAS (SAS Institute; Cary, NC, USA); significant differences ($p \leq 0.05$) among means were determined by Duncan's multiple range test (DMRT).

## RESULTS

### Characteristics of biosorbents

We found that RH had a higher specific surface area and total pore volume (4.24 $m^2\ g^{-1}$, 0.0051 $cm^3\ g^{-1}$) than RF (2.85 $m^2\ g^{-1}$, 0.0043 $cm^3\ g^{-1}$), while the average pore diameter of RH (4.824 nm) was smaller than that of RF (5.997 nm), as indicated in Table 1. The pore size distribution of RF and RH is shown in Fig. 1.

### Adsorption of RF-MeSA and RH-MeSA
#### Adsorption percentage

The adsorption percentages of both RF-MeSA and RH-MeSA increased rapidly during hours 1–6 and then attained equilibrium between hours 12 and 24. At 24 h, the adsorption percentages of RF-MeSA and RH-MeSA were 36.76% and 58.33%, respectively (Fig. 2). The adsorption percentage of RH-MeSA was significantly higher than that of RF-MeSA throughout the entire 24 h period ($p \leq 0.05$).

#### Surface morphology

The surface morphologies of RF and RH before and after MeSA adsorption (RF-MeSA and RH-MeSA) were investigated by SEM, as shown in Figs. 3A–3D. According to the

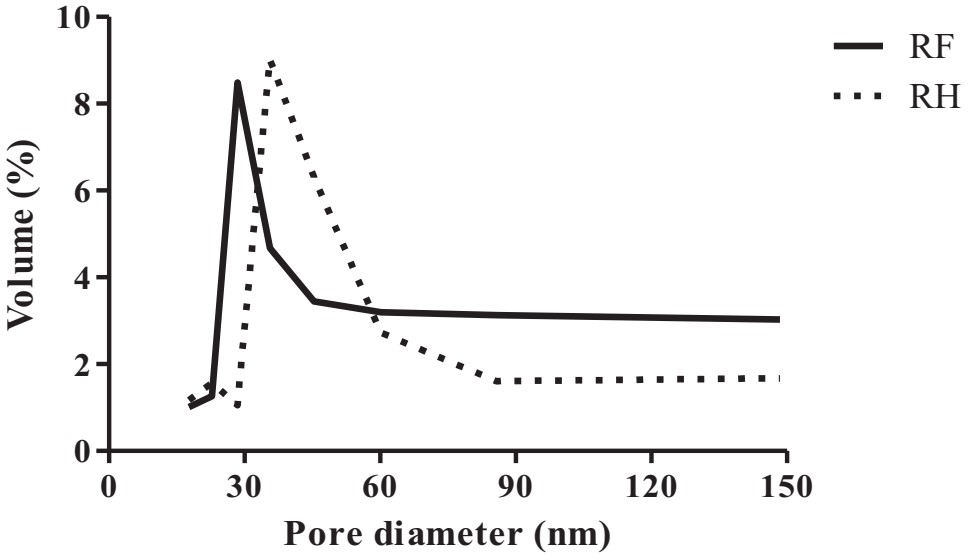

**Figure 1 Pore size distribution of rice flour (RF) and rice husk (RH).**

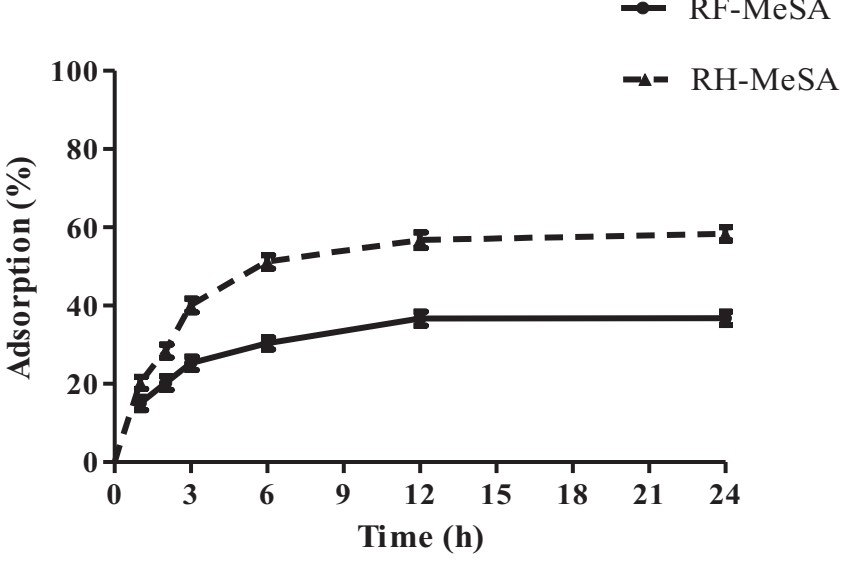

**Figure 2 The percentage adsorption of rice flour-methyl salicylate (RF-MeSA) and rice husk-methyl salicylate (RH-MeSA) at 25 °C for 24 h.**

SEM images, RF had pores and cracked surfaces on some parts of the starch granules, whereas pores were significantly reduced on RF-MeSA. In addition, RH showed a longitudinal shape with a very rough texture, while the surface of RH-MeSA was characterized by smooth channels. The EDS spectrum confirmed the MeSA adsorption on the RF and RH surfaces. The RH content showed carbon (71.11%), oxygen (26.67%) and potassium (2.22%) atoms. After MeSA adsorption, the RF-MeSA showed an increase in carbon content (84.91%) (Figs. 3E and 3F). The RH-MeSA showed a higher carbon

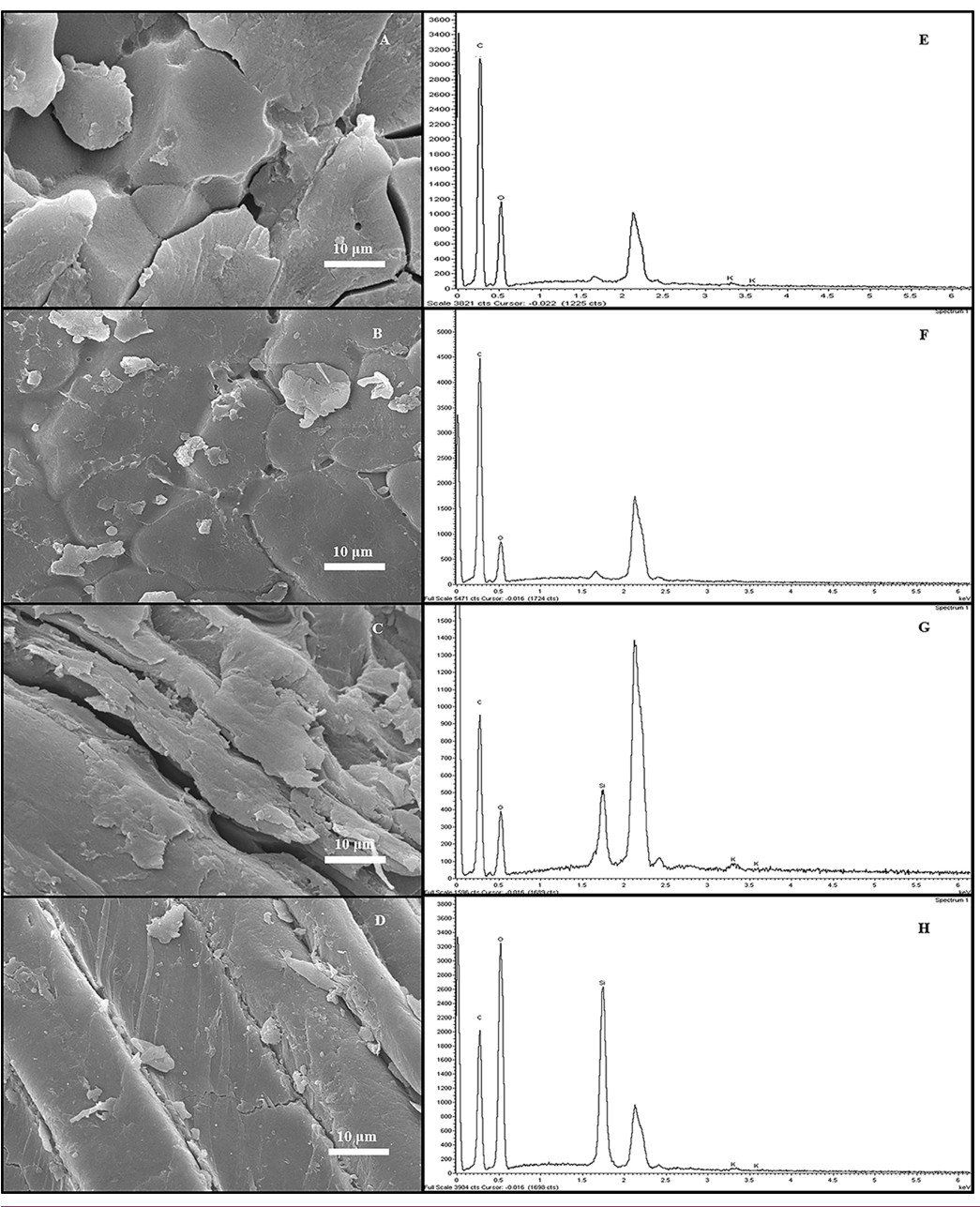

**Figure 3 SEM images and EDS spectrum of rice flour (RF) (A and E), rice flour-methyl salicylate (RF-MeSA) (B and F), rice husk (RH) (C and G), and rice husk-methyl salicylate (RH-MeSA) (D and H) at 24 h.**

(29.58%) and oxygen content (11.83%) after the adsorption process than RH (carbon = 38.46 and oxygen = 39.60%) (Figs. 3G and 3H).

## X-ray diffraction

The XRD patterns (Figs. 4A and 4B) present the structure of RF and RH before and after MeSA adsorption. The main peaks for RF occurred at 2θ diffraction angles of 15°, 17°, 19° and 23°; these peaks indicate an A-type pattern of semi-crystalline structural

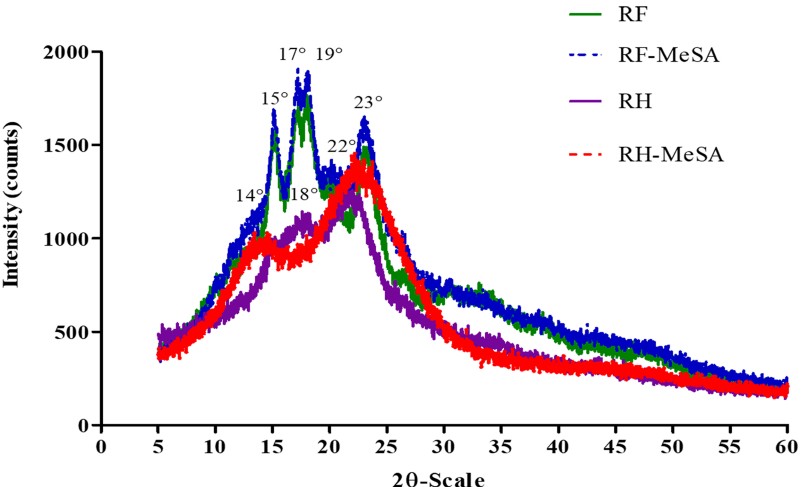

**Figure 4 XRD patterns of rice flour (RF) and rice flour-methyl salicylate (RF-MeSA) at 24 h, rice husk (RH) and rice husk-methyl salicylate RH-MeSA at 24 h.**

arrangement of the amylose and amylopectin molecules. After adsorption, the diffraction peak intensity of RF-MeSA increased at 17°, 19° and 23°, while RH-MeSA showed an increase in the peak intensity at 22° and a slight shift in peak position from 18° to 14°. The crystalline structure patterns of both RF-MeSA and RH-MeSA persisted.

### Functional groups

We used FTIR to monitor vibrational frequency changes in the functional groups of RF-MeSA and RH-MeSA, as shown in Figs. 5A and 5B. The results showed different patterns in both RF and RH before and after MeSA adsorption. Prior to adsorption, the peak at 3,460 $cm^{-1}$ for RF was assigned to the vibration of O–H stretching, and the peak at 2,900 $cm^{-1}$ was assigned to C–H stretching. The carbonyl group (C=O) of the esterified acetyl group was verified by the peak at 1,740 $cm^{-1}$, and the peak at 1,080 $cm^{-1}$ was attributed to C–OH stretching of cellulose. For RH, the adsorption peak at 3,460 $cm^{-1}$ was assigned to free hydroxyl groups present in cellulose, hemicellulose, and lignin. The C–H stretching vibration at 2,900 $cm^{-1}$ indicated the presence of an alkane functional group. The peak around 1,740 $cm^{-1}$ was assigned to C=O stretching, which can be attributed to aromatic groups in the hemicelluloses and lignin. The FTIR spectra of RF-MeSA and RH-MeSA did not show peaks at 3,460 $cm^{-1}$ (O–H stretching), but rather at 3,200 $cm^{-1}$, which corresponds with the spectrum of MeSA. The peak at 2,900 $cm^{-1}$ was reflective of C–H stretching on both RF-MeSA and RH-MeSA.

### Thermogravimetric analysis

The differences in weight loss of RF-MeSA and RH-MeSA were investigated by TGA curves, which were separated into three phases (Figs. 6A and 6B). The weight loss shown by RF within the temperature range of 40–100 °C was attributed to the loss of water; that which occurred in the temperature range of 180–350 °C corresponded to the degradation of the saccharide rings; and thermal decomposition above 400 °C

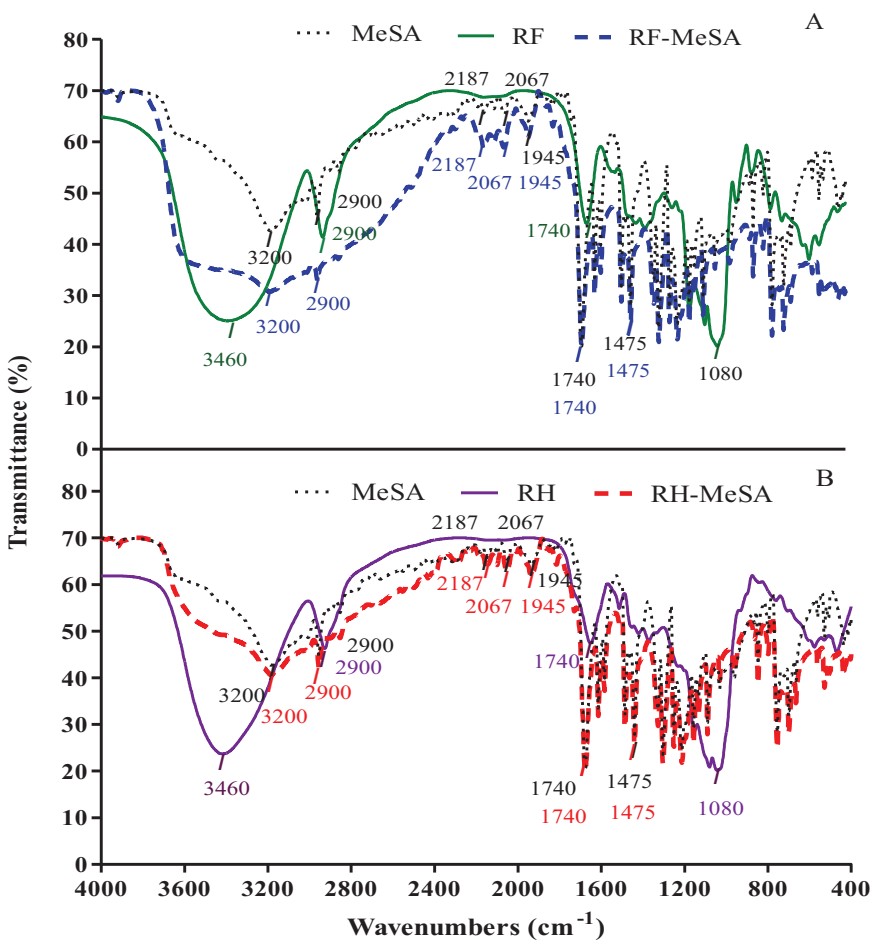

**Figure 5** FTIR spectra of rice flour-methyl salicylate (RF-MeSA) at 24 h (A), rice husk (RH) and rice husk-methyl salicylate RH-MeSA (B) at 24 h.

corresponded to char yields. In addition, the TGA curve of RF-MeSA showed two peaks in phase 2. The first peak showed thermal decomposition of the MeSA compound at around 155 °C, which resulted in a weight loss of 40.94 %. The second peak was attributed to degradation of the saccharide rings. For RH, the peak in phase 1 was caused by water evaporation. The peak in phase 2 indicated the degradation of hemicellulose and cellulose between the temperatures of 180 °C and 400 °C. The degradation of lignin occurred over a temperature range of 350–800 °C. The TGA curve of RH-MeSA showed a weight loss of 57.38%, which corresponded to thermal decomposition of the MeSA compound at 155 °C. Similarly, the TGA curve of MeSA confirmed that the weight loss that occurred at 155 °C (96.54%) could be attributed to the thermal decomposition of the MeSA compound.

## Desorption of RF-MeSA and RH-MeSA
### Effect of temperature and relative humidity on desorption
The effect of temperature on percentage of desorption was monitored for 24 h (Fig. 7A). The desorption percentages of RF-MeSA and RH-MeSA both significantly
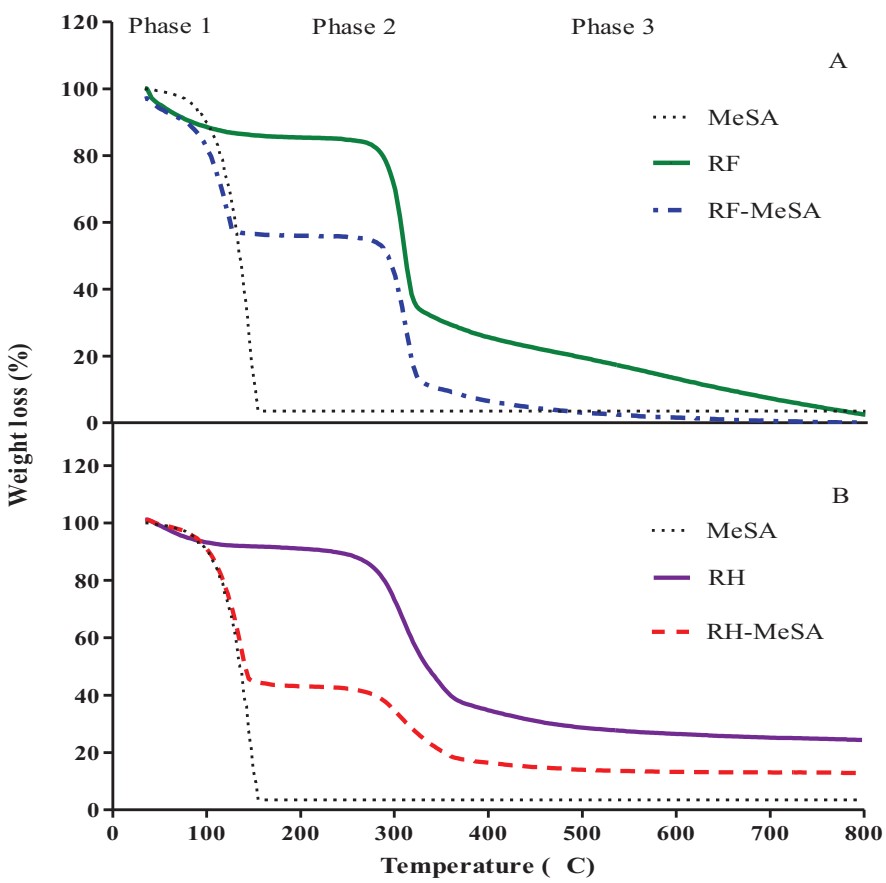

**Figure 6** TGA curve of rice flour (RF) and rice flour-methyl salicylate (RF-MeSA) at 24 h (A), rice husk (RH) and rice husk-methyl salicylate RH-MeSA (B) at 24 h.

increased with temperature increases from 25 °C to 40 °C, at 75% relative humidity ($p \leq 0.05$).

The desorption percentage of RF-MeSA and RH-MeSA, induced by relative humidity, is shown in Fig. 7B. The percentage of desorption significantly increased with increasing relative humidity ($p \leq 0.05$). At 75% relative humidity, the desorption percentage of RF-MeSA was higher than that of RH-MeSA. At 95% relative humidity; the maximum desorption percentages for RF-MeSA and RH-MeSA were 52.94% and 49.03%, respectively.

### Release kinetics

The correlation coefficient ($R^2$) and release exponent ($n$) of the zero-order, first-order, Higuchi and Korsmeyer-Peppas models of RF-MeSA and RH-MeSA are summarized in Table 2. For polymeric matrices, $n \leq 0.5$ corresponds to a Fickian diffusion mechanism, and $0.5 < n < 1$ to a non-Fickian mechanism. Table 2 shows that $n$-values less than 0.5 indicate that RF-MeSA and RH-MeSA released at different temperatures (25 °C and 40 °C, respectively) and relative humidities (75% and 90%, respectively); this finding is in line with Fickian diffusion mechanisms. Moreover, a comparison of the $R^2$ values

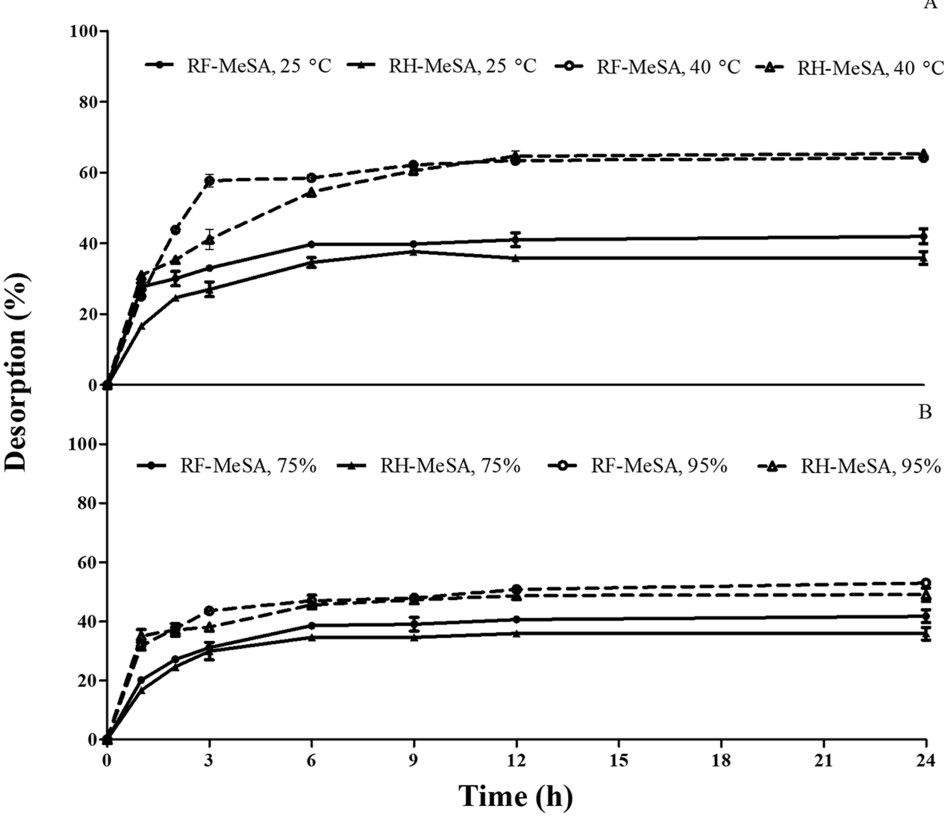

**Figure 7  The percentage desorption of rice flour-methyl salicylate (RF-MeSA) and rice husk-methyl salicylate (RH-MeSA).** (A) The percentage desorption of rice flour-methyl salicylate (RF-MeSA) and rice husk-methyl salicylate (RH-MeSA) at different temperatures (25 °C and 40 °C), relative humidity at 75%. (B) The percentage desorption of rice flour-methyl salicylate (RF-MeSA) and rice husk-methyl salicylate (RH-MeSA) at different relative humidity (75% and 95%), at 25 °C.

**Table 2  Correlation coefficient ($R^2$) according to different models and release exponent ($n$).**

| Models | RF-MeSA | | | | RH-MeSA | | | |
|---|---|---|---|---|---|---|---|---|
| | Temperature (°C) | | Relative humidity (%) | | Temperature (°C) | | Relative humidity (%) | |
| | 25 | 40 | 75 | 95 | 25 | 40 | 75 | 95 |
| Zero-order ($R^2$) | 0.4876 | 0.5539 | 0.5334 | 0.5652 | 0.5269 | 0.6992 | 0.5352 | 0.4854 |
| First-order ($R^2$) | 0.4907 | 0.5829 | 0.5382 | 0.5934 | 0.5348 | 0.7203 | 0.5434 | 0.5024 |
| Higuchi ($R^2$) | 0.7734 | 0.8128 | 0.8340 | 0.8352 | 0.8169 | 0.9065 | 0.8164 | 0.7717 |
| Korsmeyer–Peppas ($R^2$) | 0.7799 | 0.8161 | 0.7800 | 0.7194 | 0.7503 | 0.8745 | 0.7180 | 0.6951 |
| $n$ | 0.25 | 0.58 | 0.35 | 0.35 | 0.27 | 0.57 | 0.23 | 0.30 |

**Note:**
Correlation coefficient ($R^2$) according to different models and release exponent ($n$) used to describe the release of rice flour-methyl salicylate (RF-MeSA) and rice husk-methyl salicylate (RH-MeSA) at different temperatures (25 °C and 40 °C) and relative humidity (75% and 95%).

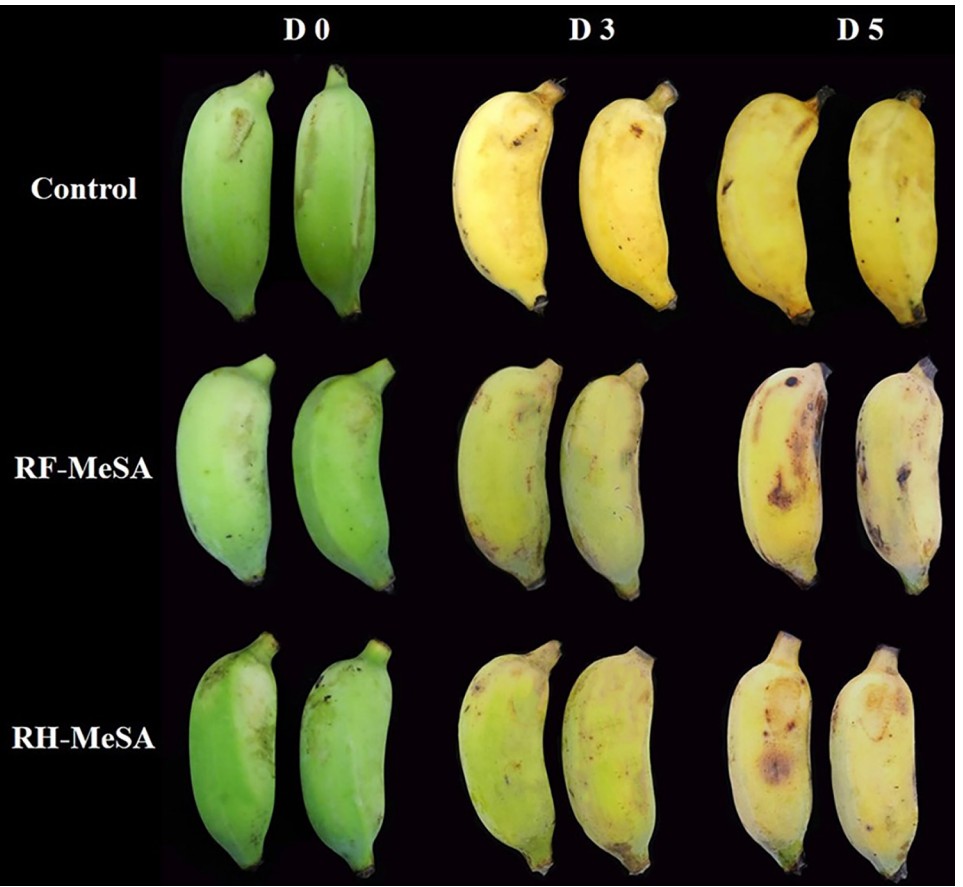

**Figure 8 Visual peel color changes of banana fruit before, during, and after the postharvest treatment with RF-MeSA, RH-MeSA and untreated fruit (control) during storage at 25 ± 2 °C.**

showed that the model best fitting the release of RF-MeSA and RH-MeSA was the Higuchi model ($R^2$: 0.7734–0.9065). These results suggest that the controlled release of MeSA can be described as a diffusion mechanism.

### Postharvest quality of banana fruit

Both RF-MeSA and RH-MeSA were applied as biosorbents to delay the ripening of bananas (Fig. 8). Bananas are a climacteric fruit and display a sharp increase in both ethylene production and respiration rate after harvest; these are two factors that accelerate ripening (*Seymour, Taylor & Tucker, 1993*). Our results showed that RH-MeSA and RF-MeSA treatments inhibited ethylene production for 1 d of storage, with the control treatment exhibiting a significantly higher value (34.72 ng $kg^{-1}$ $s^{-1}$; $p \leq 0.05$). On d 3 of storage, the climacteric peak of ethylene production in the control treatment was 1,465.28 ng $kg^{-1}$ $s^{-1}$, while those of RH-MeSA and RF-MeSA were 1,198.62 and 1,144.44 ng $kg^{-1}$ $s^{-1}$, respectively (Fig. 9A). The respiration rate, which was 1.28 μg $kg^{-1}$ $s^{-1}$ at harvest, increased in stored fruit. The respiration rate did not change significantly throughout the storage period (Fig. 7B). However, the yellowness ($b^*$ value) of the control treatment was significantly higher than that of the other treatments on d 3 and d 5 of

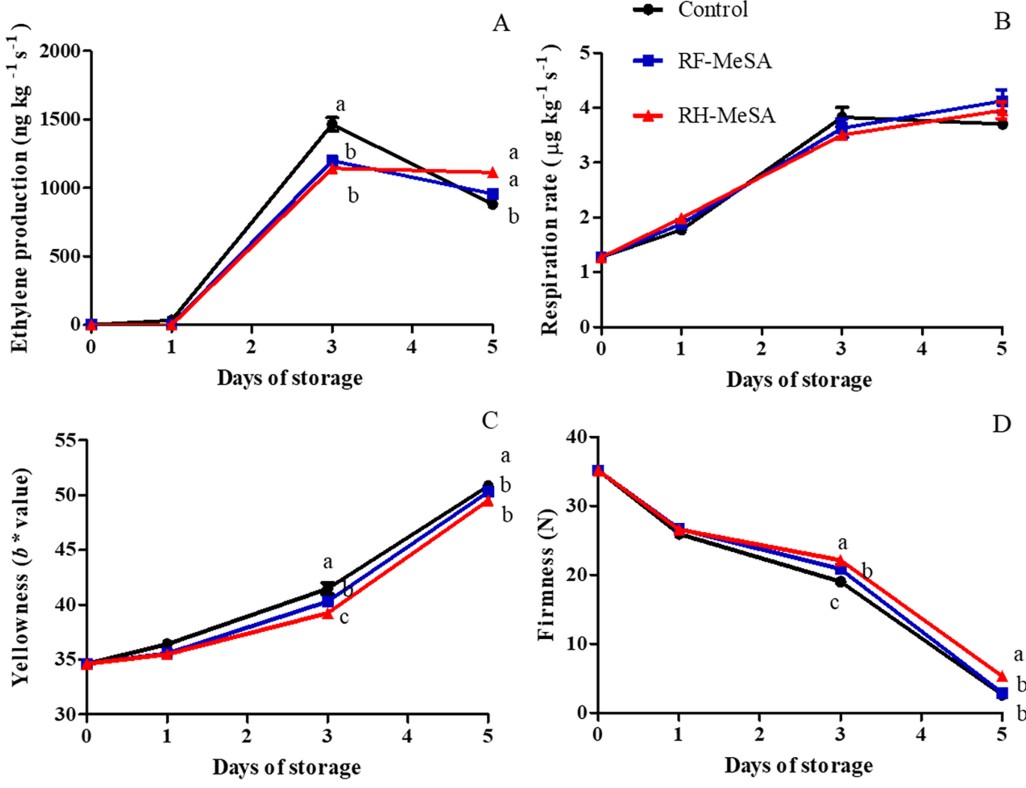

**Figure 9 Ethylene production (A), respiration rate (B), yellowness ($b^*$ value) of peel (C) and firmness (D) of 'Namwa' banana stored at 25 ± 2 °C.** Data are the mean ± SE. Different letters show significant differences ($p \leq 0.05$) for each sampling date among treatments.

**Table 3 Half-life values of the methyl salicylate (MeSA) compound to release from the rice flour (RF) and rice husk (RH) biosorbent.**

| Half-life | Biosorbents | |
|---|---|---|
| | **RF** | **RH** |
| $t_{1/2}$ (d) | 4 | 5 |

storage ($p \leq 0.05$; Fig. 9C). Similarly, the banana fruit firmness at harvest was 35.16 N and significantly decreased during storage, reaching final values of 2.62 N in control fruit and significantly higher values of 2.84 and 5.32 N in RF-MeSA and RH-MeSA treatments, respectively ($p \leq 0.05$; Fig. 7D). Moreover, the half-life release (the time taken to release 50% MeSA from the RF and RH) was 4 d and 5 d at 25 °C and 75% relative humidity, respectively (Table 3).

## DISCUSSION

These results indicate that RH has higher specific surface area and total pore volume than RF. This is likely due to the fact that the main organic compounds comprising RH are

silica, cellulose, hemicellulose, and lignin. This composition results in increased surface area and pore volume (*Ghosh & Bhattacherjee, 2013*). According to the IUPAC classification of pore size as macroporous (>50 nm), mesoporous (2.0–50 nm), and microporous (<2.0 nm), the average pore diameter of RF and RH was 5.997 nm and 4.824, respectively. Therefore, RF and RH were classified as mesoporous materials. The result can be explained because all sites on the biosorbents were available at the initial stage of the process (*Villacañas et al., 2006*), which allowed MeSA molecules to readily occupy them until the adsorption process stabilized, as reflected by the attainment of equilibrium between 12 and 24 h. However, the time required to achieve equilibrium in the adsorption process largely depends on the adsorbent structure and is influenced by the molar mass of the adsorbed compound (*Singh & Yenkie, 2006*). Removing the excess MeSA compound using Whatman® filter paper No.1 has particle retention at 98% efficiency (*Hutten, 2015*) and it did not affect the loss of active compounds that were trapped in the pores. This method has been used to estimate the removal of heavy metal ions (*Khokhotva & Waara, 2010*) and olive oil waste (*Garcia et al., 2006*). Our results showed a higher specific surface area and total pore volume for RH than for RF, which may account for the higher adsorption percentage recorded for RH-MeSA relative to RF-MeSA. The XRD of RH showed strong broad peaks of semi-crystalline structures at 2θ angle values of 18° and 22°. These peaks indicate the presence of cellulose fiber and silica in the semi-crystalline structure (*Kusbiantoro et al., 2012*). Hence, the SEM images and EDS spectrum confirmed that MeSA was adsorbed on the outer surface of RF and RH. The EDS indicated from the increased amount of carbon after MeSA adsorption on both RF and RH that MeSA was loaded on RF and RH successfully. In addition, the FTIR spectra indicated the conjugated aromatic C–C, C=O, and hydrogen bonds that were the functional groups of the MeSA compound, thereby confirming that RF and RH adsorbed the MeSA compound. The TGA implied that MeSA could be adsorbed on RF and RH, and the amount of weight loss of RF-MeSA and RH-MeSA was larger than the RF and RH samples. This result more clearly indicates that weight loss is related to thermal degradation of adsorbed MeSA molecules. RH-MeSA showed a higher weight loss than RF-MeSA, which can be attributed to the greater specific surface area and total pore volume. During the adsorption process, the MeSA molecule attaches to the active sites of the adsorbent surface and then it diffuses into the pores. Therefore, adsorption increases with the surface area and pore volume of the adsorbent (*Zhang & Blum, 2003*). In addition, the percentage of MeSA released from RH and RF increased with increasing temperature because of the weak adsorptive forces between the binding sites of the adsorbent and adsorbate (*Ofomaja & Ho, 2007*). *Chen et al. (2011)* reported that the hydrogen bonds between starch chains are broken and the crystalline region is damaged with an increase in temperature. The humidity also affects the release rate of RF-MeSA and RH-MeSA because RF and RH are hygroscopic materials (*Navaratne, 2013*). Under high humidity, RF and RH have the ability to absorb and retain water molecules from the air, leading to the displacement of MeSA molecules by water vapor (*Smith et al., 1987*). Therefore, increasing the relative humidity leads to an increase in the release of the

MeSA compound. The temperature and relative humidity are important factors for controlling the release system. In Thailand, the ambient temperatures are 25–40 °C and relative humidity is 75–85%, while the optimum relative humidity for the majority of fruit and vegetables are 90–95% (*McGregor, 1989*). Fruit and vegetables remain as living organs after harvested, and they continue to respire, leading to moisture loss via the transpiration process (*Ben-Yehoshua, 1969*). Even though, these biosorbent materials effectively delay the ripening of fruit, their application as biosorbents should be balanced against the accelerated release of MeSA from the moisture from fresh produce.

The efficiencies of RF and RH were assessed as new biosorbents for controlling the release of methyl salicylate (MeSA). In a closed system, 1 g dosages of adsorbents were investigated, showing a pattern of Fickian diffusion. Therefore, RF-MeSA and RH-MeSA at 1 g adsorbents dosage were applied to two bananas that were placed inside a PP tray. The half-life release (the time for 50% MeSA to be released from the RF and RH) was 4 and 5 d at 25 °C and 75% relative humidity. The application of RF-MeSA and RH-MeSA for delaying the ripening of "Namwa" banana fruit indicated that RH-MeSA and RF-MeSA inhibited the ethylene production of "Namwa" bananas during storage. MeSA inhibits ethylene biosynthesis by blocking the conversion of 1-aminocyclopropane-1-carboxylic acid (ACC) to ethylene (*Leslie & Romani, 1986*). Ethylene plays a major role in regulating the ripening and softening of climacteric fruit such as mangos, peaches and jujubes (*Srivastava & Dwivedi, 2000*; *Li et al., 2007*). Thus, the inhibition of ethylene production may suppress the ripening process and, thereby, maintain fruit firmness and delay color change. However, the RF-MeSA and RH-MeSA treatment had no significant effect on respiration rate. *Ding & Wang (2003)* reported that tomatoes treated with MeSA vapor (0.5 mmol $L^{-1}$) depressed their respiration rate from 18 to 6 µg $kg^{-1}$ $s^{-1}$ compared to control fruit at 20 °C. Similarly, MeSA vapor at 1.0 mmol $L^{-1}$ reduced the respiration rate (6.94 µg $kg^{-1}$ $s^{-1}$) of "Primulat" sweet cherries over 14 d, at 2 °C (*Castillo et al., 2015*). In our results, the banana fruit under both treatments showed a slight change in the peel (red orange color). The application of RH-MeSA led to a delayed ripening process compared with that of RF-MeSA; this was because the structure of RH showed greater surface area and pore volume than RF, which affected the amount of MeSA adsorbed on the surface (*Sing et al., 1985*). Moreover, a comparison of desorption percentages showed a higher overall release in RF than in RH (at 25 °C, 75% relative humidity), indicating greater retention and therefore slower release of the MeSA compound on the part of the RH biosorbent. Both RF and RH have already been used as biosorbents without modifying the surface. RF-MeSA and RH-MeSA at 1 g of adsorbent dosage promoted adsorption of the MeSA at 36.76% and 58.33%, respectively and their ability to slowly release MeSA depended upon higher temperatures and higher relative humidity. The half-life clearly confirmed the release of MeSA from the RF and RH biosorbents and the diffusion was observed to be Fickian. Because RF and RH are eco-friendly and innovative they are expected to replace the traditional adsorbents such as β-CD which are obtained from the enzymatic degradation of starch (*Lee, Dey & Lee, 2020*). MeSA inclusion with complex β-CD at 1:1 showed the highest MeSA entrapment

efficiency (59%), indicating that the MeSA release increased with increasing relative humidity and temperature (*Kant et al., 2004*). However, RF and RH have the potential to be similar to β-CD but without the need for modification.

## CONCLUSIONS

MeSA (1.0 g) was applied to RF and RH at a ratio of 2:1 (w/w; MeSA: biosorbent) at 25 °C for 24 h in order to reach equilibrium in the desorption process during postharvest treatment. Both RF and RH have the potential to adsorb the MeSA molecule, as well as to release it. RH has a higher specific surface area and total pore volume than RF, resulting in an increased capacity to adsorb MeSA. Analysis with SEM, XRD, FTIR and TGA confirmed the adsorption of MeSA on the outer surfaces of RF and RH. Moreover, the temperature and relative humidity affected the desorption percentage of RF-MeSA and RH-MeSA. Increased temperature (from 25 °C to 40 °C at a relative humidity of 75%) and increased relative humidity (from 75% to 95% at 25 °C) stimulated the release of MeSA from RH-MeSA and RF-MeSA. In addition, the kinetically controlled release of both RF-MeSA and RH-MeSA suggested that they followed a pattern of Fickian diffusion. Thus, we report for the first time that RF and RH are natural adsorbents that have potential applicability for the adsorption and controlled release of MeSA without chemical or mechanical modifications. Although the application of RF-MeSA and RH-MeSA delayed ripening of "Namwa" banana fruit, treatments at the tested concentrations induced peel disorder. Therefore, further study on pH, optimal dosage and concentration of MeSA are needed before these biosorbents are utilized for commercial treatment with bananas or other fruit.

## ACKNOWLEDGEMENTS

The authors would like to thanks Nugreen Co., Ltd. (Thailand) for partially financial support and also like to express their gratitude to Prof. Paul Holford (Western Sydney University) for his assistance in proofreading this article.

### Funding

This research work was supported by The Thailand Research Fund (PHD60I0003) and was partially funded by Nugreen Co, Ltd. There was no additional external funding received for this study. The funders had no role in study design, data collection and analysis, decision to publish, or preparation of the manuscript.

### Grant Disclosures

The following grant information was disclosed by the authors:
Thailand Research Fund: PHD60I0003.
Nugreen Co, Ltd.

### Competing Interests

The authors declare that they have no competing interests.

## Author Contributions

- Chalida Cholmaitri conceived and designed the experiments, performed the experiments, analyzed the data, performed the computation work, prepared figures and/or tables, authored or reviewed drafts of the paper, and approved the final draft.
- Apiradee Uthairatanakij conceived and designed the experiments, performed the experiments, analyzed the data, prepared figures and/or tables, authored or reviewed drafts of the paper, and approved the final draft.
- Natta Laohakunjit conceived and designed the experiments, performed the experiments, analyzed the data, prepared figures and/or tables, authored or reviewed drafts of the paper, and approved the final draft.
- Pongphen Jitareerat performed the experiments, authored or reviewed drafts of the paper, and approved the final draft.
- Withawat Mingvanish performed the experiments, authored or reviewed drafts of the paper, and approved the final draft.

## Data Availability

The raw data are available in the Supplemental Files.

## Supplemental Information

Supplemental information for this article can be found online at http://dx.doi.org/10.7717/peerj-matsci.12#supplemental-information.

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
