# Peer review of "Controlled release of methyl salicylate by biosorbents delays the ripening of banana fruit"

_PeerJ Materials Science, doi:10.7717/peerj-matsci.12_

## Round 0.1 · original submission · Minor Revisions

The comments from the reviewers, while plentiful, appear to only be seeking minor revision of your work and it would be my pleasure to consider a revised version.

Reviewer 1 ·

Basic reporting

-Your introduction needs more relevant and detail information about your project materials. I suggest that you consider some changes in first two paragraphs of your introduction and add more related background to clearly show significance of your system. for example MeSA information about formula , molar mass and non related application to your paper's goal can be replaced with more relevant MeSA application on fruits and plants.

Experimental design

-To investigate more, it is suggested to check effect of adsorbent dosage on preventing banana ripening?

-It is suggested to report the change in half time of fruit ripening.
-How experimentally author confirmed complete removal of free MeSA? just using filter paper may not be efficient enough (has author tried any chromatography methods/Centrifugation)

-Why desorption percentage increased with increasing relative humidity?

-Author need add more background information on respiration and ethylene production to make purpose of experiment clear, also as temperature (and concentration of MeSA) can have effect on respiration rates, it would be informative if author check respiration rates at different temperature and MeSA concentrations.

-To enhance the absorption capacities of developed formulation, it is suggested to modify surface of biosorbent.
-Different factors such as pH, initial concentration, sorbent dosage have effect on biosorbent, therefore considering these factors to optimize system performance by adding more experimental condition can help on performance of system.

-Page 7, Line 60 . Biosorbent materials part need more references.

-Page 9, line 105, 107. No need for abbreviation as RF and RH abbreviation has been already introduced in above part.

- Author used 2:1 mole ratio of MeSA and RF/RH and 24h incubation, so it is suggested to show the optimized mole ratio and time of reaction for both systems by reporting data for different ratios and time.

- Adsorption percentage experimental design need more explanation. E.g. author mentioned: Excess MeSA was removed from RF-MeSA and RH-MeSA samples with Whatman® filter paper No. 1 at 0, 1, 2, 3, 6, 12, and 24 h. (of what process?)

-is there any reason author tested just two temperature of 25 and 40 C and humidities of 75 % and 95 %,? more clarification and explanation would be helpful to understand the experimental design.

-What is part 2.3.1 and 2.3.2? if they are related to the mentioned reference, it is strongly recommended to add that information in the paper.

-.Data should be reported in text or table format, (not both), it is suggested to add standard deviation information as well. (Table 1)

- More characterization is recommended (e.g. EDS mapping analysis to show silicon and oxygen maps on surfaces of RH before and after adsorption)

-Page 19, Line 327 . reference is needed

-It is suggested to check postharvest quality of other fruits as well to confirm applicability of developed formulation

-Figure 1. Error bars are needed

-Page 15, line 241.Data should be reported in text or table format, (not both), it is suggested to add standard deviation as well

Validity of the findings

-Comparing you control released system ability with other biosorbent materials,such as α-cyclodextrin can help better understaning on advantages and novelity of your system.

-RF and RH have already been used as a biosorbent , so author should strongly discuss on fundamental advantage /effect of their system compared to available systems

Annotated reviews are not available for download in order to protect the identity of reviewers who chose to remain anonymous.

Reviewer 2 ·

Basic reporting

The story presented by Uthairatanakij and colleagues is clear. Further, authors made a good effort explaining their results in the discussion section. However, figures should significantly be improved for reader to fully appreciate the data authors are trying to present.

Experimental design

Experimental methodology is easy to follow and provides a good flow to the story. Further, the proposed experimental design allows for the story to be understandable by the audience.

Validity of the findings

No comment

Additional comments

The manuscript by Uthairatanakij and colleagues reports on the usage of rice husk (RH) and rice flour (RF) as bio-sorbent materials for the controlled release of methyl salicylate (MeSA). Different incubation periods of MeSA and bio-sorbent promoted absorption of the methyl ester into either RH or RF (>36%). Presence of the methyl ester within either bio-sorbent was corroborated by Fourier transform infrared spectroscopy analysis by peak overlapping of MeSA within the RF-MeSA and RH-MeSA samples. Morphology changes observed through scanning-electron micrographs were attributed the absorption of MeSA in both RF and RH. On the other hand, x-ray diffraction patterns of pristine and MeSA encapsulated RF and RH show minimal to no difference. Thermogravimetric profiles of both RF-MeSA and RH-MeSA reveal a significant mass loss (> 40%) at ~155 °C, indicative of methyl salicylate decomposition. Additionally, it was found that temperature and humidity have a significant impact on desorption of both RF-MeSA and RH-MeSA, parameters particularly relevant for the industrial application of these two bio-sorbent materials. Further, in a moisture free environment both bio-sorbent materials show to significantly delay the ripening of banana fruits.
Publication is recommended after addressing the following suggestions:
-Reviewer recommends authors to revisit the introduction, some important claims have no reference.
-Reword line 49, the idea authors are trying to convey is not clear.
-Verb conjugation in line 77 should be have instead of has.
-The reported data could be better interpreted if authors spend time improving the overall quality of reported figures.
-For example, in figure 2 remove bottom information and leave the magnification only kV values, align the letters with one another. Further, increase magnification to allow reader to appreciate the structural features discussed in the main text. This can be achieved by having a zoom-in smaller figure within each individual SEM image.
-Pore size distribution must be included to further support the absorption findings.
-Figure 3, data is hard to appreciate given the composition of the figure. Color label and combine all data into one figure Combine all data into one figure. Further, provide a control crystallinity pattern of amylose and/or amylopectin molecules.
-Data provided by figure 4 is a very big component of the story, as authors use this technique to guarantee the presence of MeSA within RF and RH. Regardless of the peak positioning discussed in the main text, authors should color label each spectrum for better interpretation of the results. FTIR spectrum of pristine MeSA lacks a stretch at approximately 2900 cm-1. Further, given the overlapping of the spectra, the fingerprint of the MeSA is impossible to interpret.
-Reviewer recommends authors to elaborate on how TGA findings correlate to the absorption data obtained. Further, sample preparation should be revisited to eliminate water loss during the ramping of the temperature.
- Reviewer invites authors to include pictures of the bananas before, during, and after the postharvest testing to visually support your findings.

---

## Round 0.2 · accepted · Accept

I look forward to seeing your article in the journal!~

Reviewer 1 ·

Basic reporting

all comments have been addressed by the author.

Experimental design

all comments have been addressed by the author.

Validity of the findings

all comments have been addressed by the author.

Additional comments

all comments have been addressed by the author.

Reviewer 2 ·

Basic reporting

No comment

Experimental design

No comment

Validity of the findings

No comment

Additional comments

No comment